# Ecological Clues to the Nature of Consciousness

**DOI:** 10.3390/e22060611

**Published:** 2020-05-30

**Authors:** Robert E. Ulanowicz

**Affiliations:** 1Department of Biology, University of Florida, Gainesville, FL 32611-8525, USA; ulan@umces.edu; Tel.: +1-(352)-392-6917; 2Chesapeake Biological Laboratory, University of Maryland Center for Environmental Science, Solomons, MD 20688-0038, USA

**Keywords:** autocatalysis, centripetality, coherence domain, consciousness, ecosystem dynamics, simultaneity

## Abstract

Some dynamics associated with consciousness are shared by other complex macroscopic living systems. For example, autocatalysis, an active agency in ecosystems, imparts to them a centripetality, the ability to attract resources that identifies the system as an agency apart from its surroundings. It is likely that autocatalysis in the central nervous system likewise gives rise to the phenomenon of selfhood, id or ego. Similarly, a coherence domain, as constituted in terms of complex bi-level coordination in ecosystems, stands as an analogy to the simultaneous access the mind has to assorted information available over different channels. The result is the feeling that various features of one’s surroundings are present to the individual all at once. Research on these phenomena in other fields may suggest empirical approaches to the study of consciousness in humans and other higher animals.

## 1. Introduction

Panpsychism is the view that all features of the natural world share elements of consciousness or mind. One need not accept this belief in its entirety to acknowledge that behaviors contributing to consciousness might be evident in natural systems seemingly removed from the brain or human mental activity, thereby contributing insights into the nature of consciousness and how it might have arisen. While physics certainly can contribute to understanding many biotic phenomena, it is limited in that its laws can treat only collections of indistinguishable objects [1]. This restriction to homogeneous variables makes it difficult to apply the fundamental laws to behaviors among radically different biological kinds [2,3]. Ecology, in contrast with physics, embraces heterogeneity in its very foundations with its emphasis on the relationships among many differing types. Some even assert that proto-ecosystem behaviors had to precede the appearance of the first organisms—that physical cycling among regions that are dominated by antagonistic reactions (e.g., oxidation/reduction) provided the crucible that fostered the first proto-organisms [4]. It is thus plausible to consider whether some features of ecosystem dynamics might resemble those of the complex phenomenon known as consciousness. There is almost universal agreement that the experience of consciousness is one of the most difficult problems facing science. Theories of consciousness are common [5,6], relying on a host of subsidiary theories and phenomena. The major challenge is to connect consciousness to theories that are amenable to quantification, and empirical investigation [7]. One such endeavor is the Global Workspace Theory, where information is considered to be held in a global workspace and distributed to brain modules associated with specific tasks [8]. Another is Integrated Information Theory [9], which is an attempt to quantify the information that is actually fed back into the neuronal system to change it. The following essay is an attempt simply to highlight two ecosystem repertoires that might contribute to these efforts to illumine the very complex phenomenon of consciousness.

The first such behavior is autocatalysis, in which major reactants and products catalyze one another in a closed fashion. Such autocatalytic dynamics are capable of engendering a progressive tendency to draw ever more resources into its active orbit. (See example below.) This phenomenon was noted in chemistry decades ago by Bertrand Russell [10], who called it “chemical imperialism” and he pointed to it as the driver behind all of evolution. Ulanowicz [11] instead used Isaac Newton’s term, “centripetality” to describe such accumulation. Despite the fact that this phenomenon is readily visible in all living systems, it appears on no one’s list of the fundamental properties of life. For example, Varela et al. [12], in their detailed pioneering description of autopoesis (as engendered by autocatalysis), do not touch upon the phenomenon. Yet, the phenomenon implicitly defines the system as a self, the focus of its component activities.

The second property concerns the simultaneity with which various sources of information are accessible to a virtual center that may be defined by centripetality. That is, at any moment, an individual’s attention may be focused on a particular thing or activity, but that agent remains conscious of a host of other sources of information—neuronal, visual, olfactory, taste or sensory—all transpiring at the same time on the periphery of the mind. The theoretical origins of such simultaneity might be traced back to physics, and in particular to the notion of a “coherence domain”. Such coherence is exhibited by persisting aggregations of hundreds or more water molecules in the plasmas of living systems [13]. Below, how the same coherent behavior might be extrapolated to work in the process of ecosystem development is discussed [14].

## 2. Towards a Conception of Self

Autocatalysis is a dynamic that appears in consciousness literature (e.g., [15,16,17]. Broadly defined, an autocatalytic set is a collection of entities, each of which benefits and is benefitted by other entities within the same set so as to maintain the configuration of the whole. To illumine some of the properties of autocatalysis, one may begin by considering simple autocatalytic cycles. An autocatalytic cycle means any cycle of processes for which each constituent process benefits the next one in the sequence [18]. In Figure 1, for example, if process A facilitates another process, B, and B benefits C, which in its turn augments A, then the activity of A indirectly promotes itself. The same goes, of course, for B and C.

As an ecological example of autocatalysis, one may consider the aquatic community that develops around a family of aquatic weeds known as Bladderworts (genus *Utricularia*) [19]. Scattered along the feather-like stems and leaves of these plants are small visible bladders that normally maintain within themselves a lower osmotic pressure than their surroundings (Figure 2a). At the end of each bladder are a few hair-like triggers, which, when touched by any tiny suspended animals (such as 0.1 mm water fleas), will open the end to suck in the animal (Figure 2b). It then closes the opening and the entrapped animal becomes food for the plant (akin to what happens with the more familiar Venus fly trap).

In nature, the surface of Bladderworts always hosts the growth of a nutrient-rich algal film. This surface growth serves as ready food for a variety of microscopic animals. The algae attached to the leaves benefit from the stream of nutrients that water currents bring to them, in contrast with free-floating plankton, which quickly exhaust nutrients from a small quiescent surrounding sphere [20]. Hence, bladderworts provide a surface upon which the algae can grow; the algae feed the micro animals, which close the cycle by becoming food for the Bladderwort (Figure 3). Bladderwort communities dominate many nutrient-poor freshwater habitats, like the inner reaches of the Florida Everglades grasslands.

The autocatalytic dynamics among the *Utricularia* plant, U, the attached algae (periphyton) film, P, and the zooplankton density, Z, can be modeled after Ulanowicz [19] in abbreviated form (The reader is referred to Ulanowicz [19], where the dynamical equations and their steady-state solutions are more completely articulated. Limits, such as nutrient scarcity and periphyton shading, were eliminated here for simplicity, leaving autocatalysis unbounded.) as:(1)$$\frac{dU}{dt}=\alpha UN+\varphi ZU-mU$$
(2)$$\frac{dP}{dt}=\gamma PN-\rho ZP-\mu P$$
(3)$$\frac{dZ}{dt}=\rho ZP+\sigma Z\Psi -\varphi ZU-\xi Z$$
(4)$$\frac{d\Psi }{dt}=\eta \Psi N-\sigma Z\Psi -\varepsilon \Psi$$
where *α, γ, φ,*
ρ*, σ* and *η* are Lotka–Volterra reaction constants, *N* is the concentration of dissolved nutrient, Ψ the density of free-floating phytoplankton, and *m,*
μ*,*
ξ and *ε* are the mortality rates of *U*, *P*, *Z* and Ψ, respectively.

If one identifies *M* = *U* + *P* + *Z* as the total mass of the *Utricularia* system, then adding the first three equations yields,
(5)$$\frac{dM}{dt}=\alpha UN+\gamma PN+\sigma Z\Psi -\mathrm{mortalities}$$

Now, if algae could not anchor to the *Utricularia* (i.e., *P* = 0), there would still be a slight subsidy to the *Utricularia* system via its carnivory in the amount σZΨ. This advantage would be almost negligible, however, because of the nutrient limitation to free-floating phytoplankton mentioned above. By attaching to a leaf surface, however, nutrients are swept close to its surface, significantly accelerating absorption [21]. Thereby, the system, *M*, accretes by γPN, which in turn increases the rate of zooplankton growth by ρZP, and the assist to *Utricularia*, φZU inflates accordingly.

Ulanowicz [19] did not numerically integrate this set of equations. Rather, he set all derivatives to zero and solved for the steady-state values of *U*, *P* and *Z*. After obtaining realistic estimates for all parameters, it was concluded that the feedback, φZU adds roughly 33% to intrinsic *Utricularia* growth, αUN allowing it to outcompete other systems when nutrients are low.

While autocatalysis among simple, invariant chemicals remains strictly mechanical, whenever the reactants are malleable to an extent, or when contingencies occur in, or impinge upon, elements of the cycle, endogenous (within system) selection can occur. For example, in the *Utricularia* system, if there happens to be some contingent change in the surface algae that either allows more algae to grow on the same surface of Bladderwort (e.g., by becoming more transparent), then the effect of the increased algal activity induced by that contingent event will be rewarded two steps later by more Bladderwort surface. The activities of all the members of the triad will be increased. Conversely, if the change either decreases the possible algal density or makes the algae less palatable to the microanimals, then the rates of all three processes will be attenuated. Simply put, contingencies that facilitate any component process will be reinforced, whereas those that interfere with facilitation anywhere will be decremented. The net result is the facilitation (selection) of features of participating members and pathways over and above non-participating entities. Evidence of such selection in the *Utricularia* system include the plant morphology of hair-like leaves that provide a maximal surface to biomass ratio to enable it to host more periphyton. Additionally, some subspecies exude mucilage to help the algae adhere to its surface, while others, in low-pH waters, host more transparent periphyton.

Having established how autocatalysis can select for changes that contribute to greater autocatalysis, it is a minor step to broaden the narrative to encompass exchanges with the rest of the world. Processes A, B and C cannot occur without inputs of energy or resources from elsewhere. So if a contingency in B happens to enable it to import more needed resources into itself, that change will be reinforced. If the change decreases the ability of B to import resources, autocatalysis will regress, thereby decrementing the change. Now, the same dynamic applies not only to B, but to all members of the cycle. Hence, over time, the interaction of autocatalysis with contingent changes induces the system to import ever greater amounts of necessary resources over all possible inputs to the system. Such “centripetality” is illustrated schematically in Figure 4.

It is not possible to completely mathematize the action of centripetality beyond the simple representation of positive feedback, as was carried out above for the *Utricularia* system. One notes that the extent of the autocatalytic system, *M,* is inflated by γPN+σZΨ, thereby extracting nutrients from the stores of *N* and competing phytoplankton, Ψ. More generally, however, centripetality arises through the boundary conditions, which are dominated by contingencies. Kauffman [21] characterizes those boundary constraints and events as the “adjacent possible” and argues how such contingencies cannot even be categorized in advance. His simple example is to challenge anyone to list all the uses of a screwdriver. One begins by drawing up a list of the most obvious alternative uses, but discovers that each new use suggests further variations, progressively lengthening the list ad-infinitum. He concludes generally, therefore, that boundary conditions remain “unprestatable”.

The centripetal system in Figure 4 is capable of exerting agency upon the external world—it acts as an entity upon its surroundings so as to aggrandize itself. In fact, if two autocatalytic systems inhabit a region with finite resources, competition will inevitably ensue. Heuristic evidence of centripetality exerted by *Utricularia* systems lies in the observations that *Utricularia* usually forms uniform stands in relatively clear water by sequestering resources to the exclusion of other competitors, such as phytoplankton. Hence, competition as a process is always secondary and *subsidiary* to the more primary mutual beneficence at the next level down. The inward radial directions of inputs in Figure 4 imply a virtual *center* towards which action is directed. Within the brain and body of a higher organism, such directionality engendered by autocatalytically related sets of neurons can support the notion of self, id or ego—central players in the phenomenon of consciousness.

**Hypothesis** **1.**
*Autocatalysis among the processes of the brain works to produce an image of the self as the focus of all mental activities.*


## 3. Bringing the Outside World Within

The foregoing description of centripitality tends to gloss over considerations of the rates of signal transmission and phase relationships among them. For example, the material in each link in an ecosystem trophic web may take days, weeks or years to travel from one element to the next, yet the response of the system sometimes must take place on a much shorter timescale. How is this possible?

A key observation in this regard was made by physicists studying interstitial water in biological plasmas [13]. It appears that collections of many water molecules persist as quasi-stable aggregations within living systems. These “coherence domains” were capable of catalyzing necessary molecular biological processes, such as photosynthesis. Now, this phenomenon was strange, because the electromagnetic (EM) forces among the molecules are capable, at biotic temperatures, of maintaining coherence only over a distance of a few molecules. How does the aggregation persist in the face of molecular perturbations? Physicists theorized that the phase correlations within the larger coherent ensemble were maintained not by their EM fields directly (which travel near the speed of light), but by their potentials that can travel across the quantum vacuum at a phase velocity that can exceed the speed of light [22].

Quantum esoterica aside, for a coherence domain to persist, simultaneous activities at two separate hierarchical levels are required: slower and stronger forces govern the movements of the water atoms themselves, but weaker and faster communication keeps those movements in coherent cadence at longer distances. This complex dynamic bears analogy to the movement of an Olympic rowing scull. The scull is propelled by the exertions of eight strong rowers, who are kept in synchrony by a diminutive coxswain, who calls out the cadence.

At the macroscopic scales of an ecosystem, bulk mass and energy flows are maintained among species over pathways that are relatively slow. The coordination and phasing of autocatalytic activities are maintained by much faster communication routes, such as visual cues (speed of light), sound, olfactory signals and pressure waves (in water). Such biosemiotic cues co-evolve with autocatalytic trophic activities that constitute the bulk flows to create what has been called “biosemiotic scaffolding” [23], which maintains the flow network in existence. That is, the complex of rapid biosemiotic signals develop so as to coordinate and support the slower trophic transfers in autocatalytic, self-re-enforcing fashion [24].

Such bi-level, parallel patterns of coordination likely arise in other domains of ordered phenomena, such as neuronal activities in the brain (ibid.). As is well-known, synaptic firings are relatively slow, taking on the order of 0.1 s to transpire. Brain activities, however, do not appear to be dominated by transmission along concatenated neuronal pathways, but instead as spatio-temporal patterns of neuronal bulk firings. It would appear plausible that these larger patterns of firings might be coordinated by faint electromagnetic emissions emanating almost at the speed of light from other bulk firings. That is, the stronger, slower firings are being coordinated by weaker but much faster EM pulses that serve as a biosemiotic scaffold.

The extreme speed with which the weak EM pulses travel means that information received at disparate regions of the central nervous system could immediately be sensed over all the approximately 20 cm domain of the brain. This could likely give rise to the sensation of simultaneity, whereby one has the feeling that information of various sorts is accessible all at once, allowing one to direct attention to the source of greatest immediate import. The dynamic is similar in effect to the Global Workspace Theory, except the workspace here is physically distributed, rather than central (although with respect to the configuration of processes, all incoming signals point towards a virtual center [i.e., the “self”, as in Figure 4]).

**Hypothesis** **2.**
*The sensation in consciousness that various sources of information are simultaneously available is a manifestation of a coherence domain existing among all sensory regions of the brain.*


## 4. Centripetality and Simultaneity as Concerns Empirical Projects

Although centripetality and simultaneity are, at best, cryptic in the two leading theories (the Global Workspace Theory and the Integrated Information Theory), their expositions here suggest possible new pathways for studying and quantifying these phenomena.

Tononi [25], for example, quantifies his integrated information using the Kullback–Leibler Divergence formula to quantify change in before and after probability distributions. This is a legitimate way of reckoning information, but it only measures differences in marginal probabilities. The brain and its mental activities are more fully conceived as networks of relationships among entities or events [26]. If one can identify and quantify the connections that embody these relationships, one can array them in matrix form, which reveals both the topology of the interactions and the relative magnitudes of each one (Figure 5). As is preliminary to the derivation of the Kullback–Leibler Divergence, one first identifies the diversity among those flows as,
(6)$$H=-\sum_{i.j}p_{ij}log\left(p_{ij}\right)=-\sum_{i,j}\frac{T_{ij}}{T..}log\left(\frac{T_{ij}}{T..}\right)$$
where $p_{ij}$ is the joint probability of the flow from *i* to *j,*
$T_{ij}$ is the actual value of the *ij*th relationship and a dot in place of an index indicates summation over that index. Therefore T. is the sum of all system flows.

*H* is conventionally and mistakenly referred to as the “entropy” of the distribution of relationships, when in fact it can be decomposed into two complementary non-negative components,
(7)$$H=+\sum_{i,j}\frac{T_{ij}}{T..}log\left(\frac{T_{ij}T..}{T_{i.}T_{.j}}\right)-\sum_{i,j}\frac{T_{ij}}{T..}log\left(\frac{T_{ij}^{2}}{T_{i.}T_{.j}}\right),$$
orH=A+Φ

In information theory, the first term is called the “average mutual information”, *A* (≥0) and it quantifies the degree of constraints that bind the relationships into a whole system, whereas the second term is called the “conditional entropy”, *Φ* (≥0), which measures the apophatic lack of constraint, or freedom remaining among the relations [27]. Thus, one sees that, more generally, H represents not just entropy (unconnectedness or freedom), but information on binding as well [28]. (It should be mentioned in passing that autocatalysis tends to contribute to greater and more efficient streamlining of the network [higher A]).

Ulanowicz [27] applied this analysis of conditional probabilities to networks of trophic exchanges in ecosystems, which in principle can be assessed directly. The relative values of A and Φ associated with any particular ecosystem indicate how streamlined (organized) or how branched (indeterminate) its network is, respectively. The compilation of ecosystem networks from various habitats and conditions reveal a surprisingly narrow grouping of the relative values of A and Φ [29], clustering around a value of A/H ≈ 0.40. This observation inferred that ecosystems do not perform at maximum (streamlined) efficiency, but rather retain a large degree of indeterminacy that can serve as a wellspring for new configurations capable of resisting or accommodating novel threats.

It is well accepted that brain activity, likewise, is neither very rigid (organized) nor too chaotic [30]. A question of great interest might be whether networks of neural activities exhibit clustering similar to that of ecosystems and where along the spectrum of organization might such cluster lie. One experiment that immediately presents itself is whether there is any significant change in the relative values of A and Φ between conscious and unconscious brain activity.

The system-level information indices are also useful in ecosystems to identify which links are most important or limiting [31]. Such knowledge might also be useful for assessing what is important for consciousness by comparing the differences in limiting transfers between states of consciousness and unconsciousness.

Other tools from network analysis could also be applied to neuronal networks. The autocatalysis present in networks often is associated with (pathway) cycles in the activity network. Provided the network is of manageable dimension, all simple cycles can be identified and abstracted from the starting network [32]. Comparing these patterns of cycles in conscious and unconscious states could identify those loops most important in autocatalytic and centripetal phenomena. The list of other possible methods that can be borrowed from ecological network analysis is wider still and involves various measures of indirect effects, pathway distances of causal chains (akin to trophic analysis in ecology) and the propagation of positive and negative influences [33].

## 5. Coherence Phenomena

The empirical investigation of coherence phenomena in the brain is confounded by the method used to measure brain activity. That is, brain activities are usually mapped by sensing the EM emissions of synaptical firings. The method of measurement thus coincides with the very phenomenon hypothesized to maintain coherence. How to deconvolute the neuronal firings from the phenomenon that both signals the existence of a firing and coheres multiple firings with one another seems to be a problem almost as complex as observing consciousness itself. Nevertheless, enterprising and brilliant electrical engineers could possibly observe coherence signaling using something akin to “stereoscopic” sensors of separate brain scanners.

There then remains the problem of how an EM signal could trigger a firing and how different EM signals emanating from different locations in the brain could be distinguished one from another.

There remains yet two other possibilities. The coherence signal might possibly be other than EM emissions from firing neurons. Mae-Wan Ho [34], for example, wrote about how living tissue is close to chemical equilibrium and that disequilibria can propagate through the system at a very rapid rate. That rate need only be fast in comparison with neuronal firings, but could remain slower than the almost-as-fast-as-light EM signals. The technical problem would remain how to detect and measure the development of such intercellular communications. Finally, Stuart Kauffman [17] hypothesized that neuronal activities might be synchronized via quantum entanglement.

## 6. Conclusions

The conscious experience of identity is likely a neuronal phenomenon analogous to the centripetality, or inward-directedness, that is engendered by autocatalytic cycles of processes, such as the characterization of dynamics in ecosystems and numerous other ensemble entities. If neuronal activity can be characterized in terms of networks among portions of the nervous system, then autocatalytic activity can likely be tracked using information metrics such as average mutual information to quantify differences between conscious and unconscious brain activities.

The apparent simultaneous access to different sorts of information in the mind is very likely related to coherence phenomena within the brain. The slower propagation of serial neuronal firings is likely coordinated by faster but weaker modes of communication facilitated by EM emissions from the synaptic firings themselves, by quantum entanglement or by metabolic waves that can race through tissues.

The fact that both aspects of consciousness exhibit dynamics similar to those in macroscopic systems, like ecological communities, shows promise that the recondite problem of consciousness may soon be partially resolved through the study of large-scale systems.

## Figures and Tables

**Figure 1 entropy-22-00611-f001:**
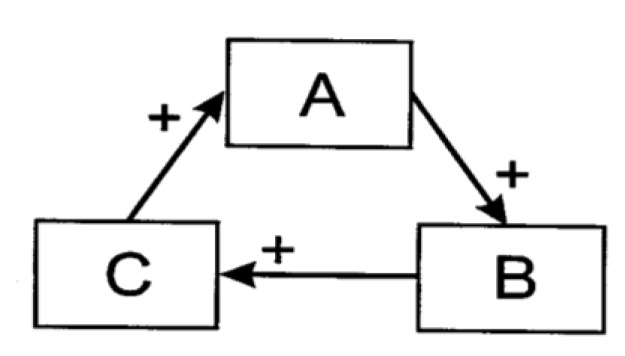
Schematic of a hypothetical three-component autocatalytic cycle.

**Figure 2 entropy-22-00611-f002:**
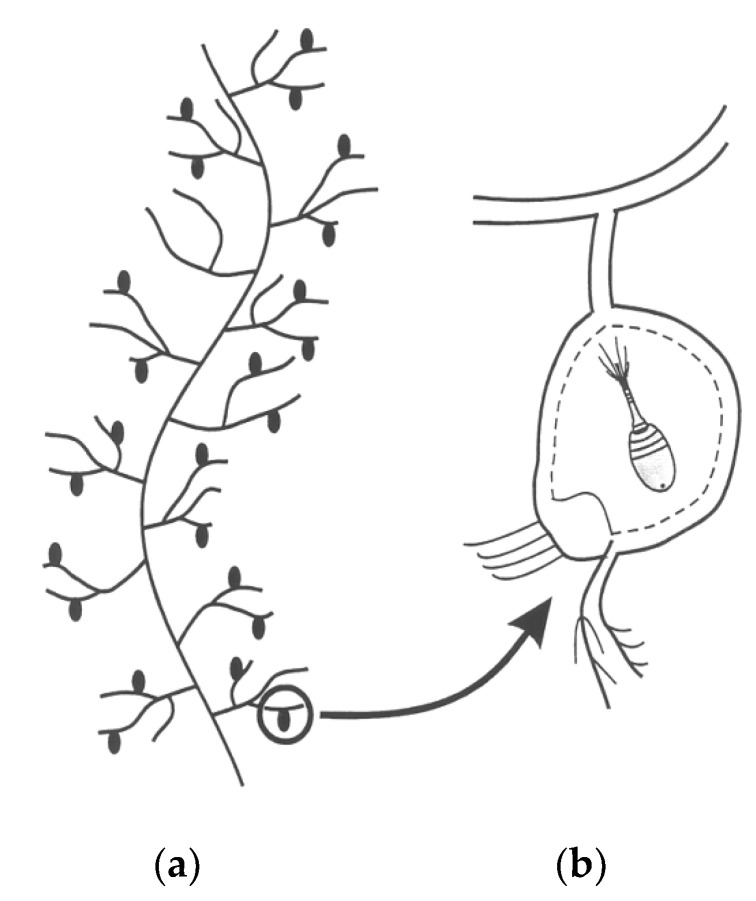
(**a**) Sketch of a typical “leaf” of *Utricularia floridana*, with (**b**) detail of the interior of an utricle containing a captured invertebrate.

**Figure 3 entropy-22-00611-f003:**
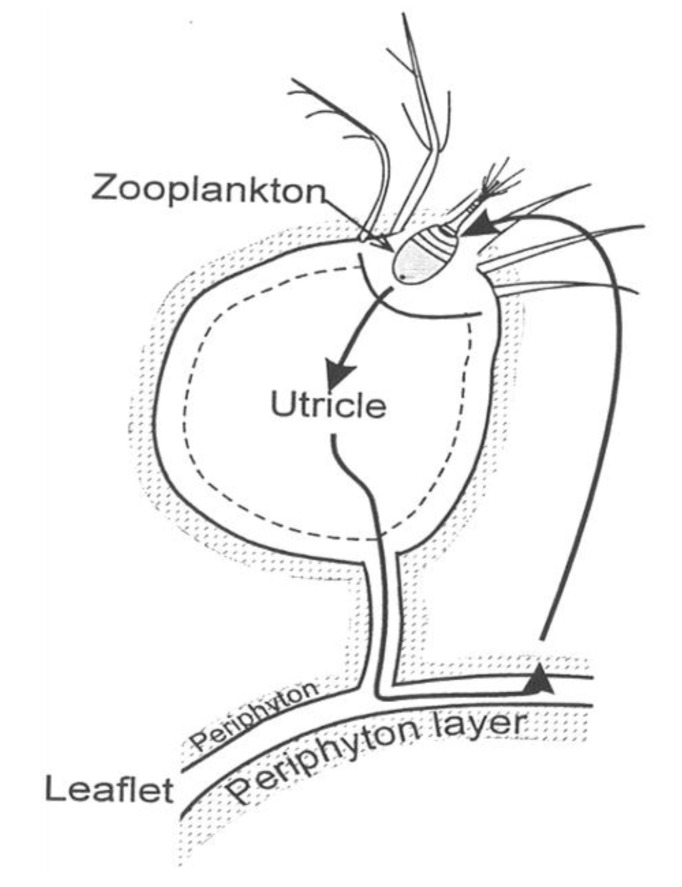
Schematic of the autocatalytic loop in the *Utricularia* system. The *Utricularia* plant provides necessary surface upon which periphyton (an algae film designated by the speckled area) can grow. Zooplankton consumes periphyton and is itself trapped in bladder and absorbed in turn by the *Utricularia*.

**Figure 4 entropy-22-00611-f004:**
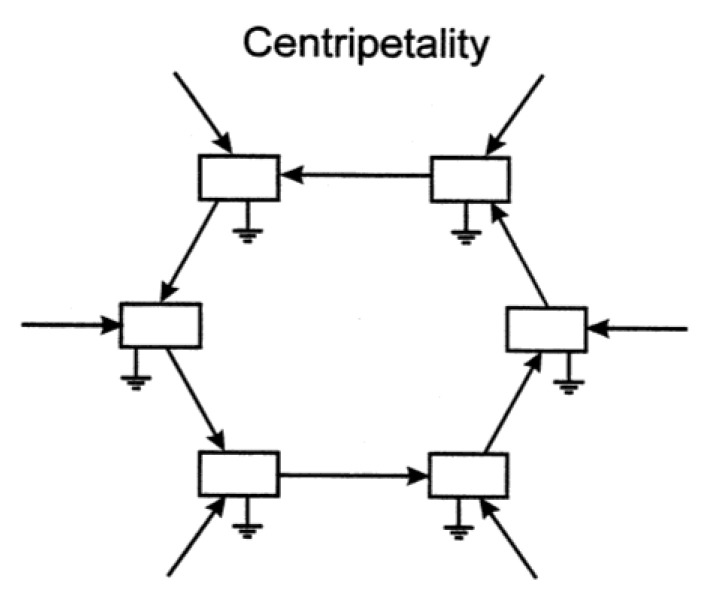
Centripetal action as engendered by autocatalysis.

**Figure 5 entropy-22-00611-f005:**
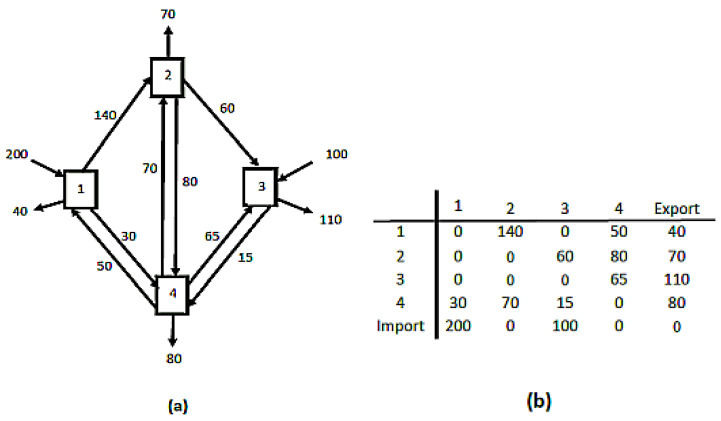
(**a**) A hypothetical four-node weighted network. (**b**) Network (**a**) arrayed as a matrix. (Example: $T_{23}=60)$.

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
