# Peer review of "Ecological Clues to the Nature of Consciousness"

_entropy, 2020, doi:10.3390/e22060611_

Round 1

Reviewer 1 Report

A very important article, which explores material (e.g. chemical, biological, biophysical) aspect of consciousness, which is a refreshing extension from merely information-theoretic approaches predominant in early AI. This is consistent with approaches in AI by A. Sloman and others. The article, although based on longitudinal research, may be a part of the trend that is just brewing. Mathematical grounding is a plus. Some methodological section/s may be helpful but not required. We will be lucky to publish this excellent work.

Odd formatting with different font sizes (one paper dated 1000 years ago mistakenly and so on -- require careful proofreading by our staff.

Author Response

I thank Reviewer #1 for his/her encouraging appraisal! I added a citation by Sloman to the references. The reviewer signed the review, but his/her name was not available to me. If she/he were willing, I would like to add their name to my acknowledgements.

Reviewer 2 Report

The author proposes a fascinating hypothesis that might even be correct or worthy of more research. The writing style is much too terse. The author assumes familiarity with the many references he makes. As I have not read these papers I could not follow his arguments. Based on my past readings of the author I have a high opinion of his scholarship and found his past works easy to follow. I am therefore not in a position to really judge the merits of his proposal. I found answering the questions above almost impossible because I could not follow his arguments. He needs to spend some time summarizing the arguments in the papers he references. To do justice to his argument I would have to read almost all the papers he referenced. Perhaps by admitting my difficulties in following his arguments I have disqualified myself as a reviewer but I believe it reveals that this paper needs a major rewrite. I was excited to read this paper because of my high opinion of the author and the fascinating hypothesis he proposed. I hope that the author takes my suggestions to heart and rewrites the paper, which I would be happy to re-review. Even if you do not want retain me as a reviewer of the revised paper please send it to me any way as I am fascinated by Ulanowicz's hypothesis. 

Here are 2 small points for the author:

  1. The author should reference the auto-catalytic work of Stuart Kauffman in addition to Varela, F.G., Maturana, H.R. and Uribe
  2. listing olfactory and smell in the same list is redundant

A double blind review was not possible given the author's many self references.

Author Response

I thank the reviewer for expressing his/her encouraging confidence in my work!

I understand the reviewer’s frustration over the compact nature of the article. I saw this as an exploratory trial balloon or note tying together ecology and consciousness theories. My intention was to mention the related phenomena and ideas in passing and to provide an appropriate citation for those who might want to study the topic in more detail. Although I didn’t rewrite the text in its entirety, I did examine it sentence by sentence and encountered ambiguous phraseology that might have confused the reader. My many emendations appear in red on the submitted ms. I hope it reads a little clearer. If the reviewer still sees points that need amplification or clarification, I would be willing to reconsider and expand on those items.

I have added a citation to Stu Kauffman’s work on consciousness.

I have eliminated the redundancy olfactory/smell from the text.